# Benzo[A]Pyrene Biodegradation by Multiple and Individual Mesophilic Bacteria under Axenic Conditions and in Soil Samples

**DOI:** 10.3390/ijerph20031855

**Published:** 2023-01-19

**Authors:** Alexis Nzila, Musa M. Musa, Emmanuel Afuecheta, Assad Al-Thukair, Saravanan Sankaran, Lei Xiang, Qing X. Li

**Affiliations:** 1Department of Bioengineering, King Fahd University of Petroleum and Minerals, Dhahran 31261, Saudi Arabia; 2Interdisciplinary Research Center for Membranes and Water Security, King Fahd University of Petroleum and Minerals, Dhahran 31261, Saudi Arabia; 3Department of Chemistry, King Fahd University of Petroleum and Minerals, Dhahran 31261, Saudi Arabia; 4Interdisciplinary Research Center for Refining and Advanced Chemicals, King Fahd University of Petroleum and Minerals, Dhahran 31261, Saudi Arabia; 5Departments of Mathematics, King Fahd University of Petroleum and Minerals, Dhahran 31261, Saudi Arabia; 6Guangdong Provincial Research Center for Environment Pollution Control and Remediation Materials, College of Life Science and Technology, Jinan University, Guangzhou 510632, China; 7Department of Molecular Biosciences and Bioengineering, University of Hawaii at Manoa, Honolulu, HI 96822, USA

**Keywords:** bioremediation, polyaromatic hydrocarbons, bacterial consortia, chromatography, mass spectrometry

## Abstract

To date, only a handful of bacterial strains that can independently degrade and utilize benzo[a]pyrene (BaP) as the sole carbon source has been isolated and characterized. Here, three new bacterial strains—JBZ1A, JBZ2B, and JBZ5E—were isolated from contaminated soil and, using 16S rRNA sequencing, were identified as *Brad rhizobium japonicum*, *Micrococcus luteus*, and *Bacillus cereus*, respectively. The growth ability of each individual strain and a consortium of all strains in the presence of BaP (4–400 µmol·L^−1^, pH 7, 37 °C) was identified by the doubling time (dt). The results illustrate that dt decreased with increasing BaP concentrations for individual strains and the consortium. The optimum growth conditions of the consortium were 37 °C, 0.5% NaCl (*w*/*v*), and pH 7. Under these conditions, the degradation rate was 1.06 µmol·L^−1^·day^−1^, whereas that of individual strains ranged from 0.9 to 0.38 µmol·L^−1^·day^−1^. *B. cereus* had the strongest contribution to the consortium’s activity, with a degradation rate of 0.9 µmol·L^−1^·day^−1^. The consortium could also remove BaP spiked with soil but at a lower rate (0.01 µmol L^−1^.day^−1^). High-performance liquid chromatography–high-resolution tandem mass spectrometry permitted the detection of the metabolites of these strains, and a biodegradation pathway is proposed.

## 1. Introduction

Benzo[a]pyrene (BaP) is one of the most important polycyclic aromatic hydrocarbons (PAHs) present in the environment. Similar to other PAHs, BaP is derived from petroleum products; thus, it is released in the environment through oil exploitation, transport, and storage, as well as through the exhaust of automobile engines [1]. This compound is also found in coal tar and tobacco, and it is generated by the incomplete combustion of organic matter, generally over a temperature range of 300–600 °C, explaining its presence in grilled meat [2,3]. Consequently, BaP is a ubiquitous pollutant, and this is worsened by its extremely low solubility in aqueous media, leading to its accumulation and associated toxicity in the environment, animals, and humans [1]. BaP is a known carcinogen, and its mode of action stems from its biotransformation to dihydroxy-epoxy-tetrahydro-BaP, which binds to DNA (forming DNA adducts), leading to errors in DNA replication and, eventually, uncontrolled cell division or cancer [4,5]. BaP also has negative effects on the development and function of the immune system and fertility in offspring; these findings have also been reported in animal models [4,5,6]. Thus, the removal of BaP from the environment remains an important priority, and biodegradation, a process that rests on the ability of microbes to utilize pollutants as sources of carbon and energy, remains an attractive strategy to remove these pollutants from the environment because this approach is cost-effective and environmentally benign.

PAHs can be categorized as low-molecular-weight compounds (LMW-PAHs) with two or three rings or high-molecular-weight compounds (HMW-PAHs) with four or more rings, such as pyrene (four rings) and BaP (five rings). The literature is replete with reports describing the bacterial degradation of LMW-PAHs and, to a certain extent, of pyrene [7,8,9,10]. Comparatively, limited work has been conducted on BaP biodegradation, owing to its recalcitrance to biodegradation due to the stability of the conjugated double-bonds in its fused rings [7]. The degradation of BaP under mesophilic conditions has been reported using *Beijerinckia* sp. B-836, *Mycobacterium* sp. PYR-1, *Sphingomonas paucimobilis*, *Burkholderia cepacia*, and *Pelomonas saccharophila* [11]; however, careful observation of these reports revealed that BaP biodegradation was achieved with the use of BaP along with a growth substrate (a substrate that can be easily degraded), such as glucose, salicylic acid, naphthalene, phenanthrene, or biphenyl. This is a classic case of cometabolism [11]. To the best of our knowledge, only four bacterial species have been revealed to degrade BaP as the sole source carbon as single strains (*Bacillus subtilis*, *Zoogloea* sp., *Bacillus subtilis*, and *Ochrobactrum* sp.) [11]. Other reports provided evidence of BaP biodegradation, albeit in the context of bacterial consortia (two or more different strains or species) [11].

In this study, we investigated the ability of three bacterial species to degrade BaP. Individually, each of these strains were found to be capable of actively degrading BaP, and this degradation was even more pronounced when these bacteria were used as a consortium. The BaP metabolites generated by these strains were identified, and the biochemical pathway of degradation was investigated using high-performance liquid chromatography–high-resolution tandem mass spectrometry (HPLC-MS/MS). Furthermore, the ability of these bacteria to degrade BaP in spiked soil samples was also investigated.

## 2. Materials and Methods

PAHs (BaP, pyrene, anthracene, phenanthrene, and naphthalene), monoaromatics (catechol, phthalic acid, and salicylic acid), and all chemicals used in the preparation of the culture media were purchased from Sigma-Aldrich (St. Louis, MO, USA). The purity of BaP was ≥96%, that of PAHs and monoaromatics was >98%, and that of the other chemicals and solvents used in this study was ≥97%. Chemicals and solvents used for gas chromatography (GC) and liquid chromatography were of analytical grade. Luria–Bertani (LB) medium was purchased from Difco (Detroit, MI, USA).

### 2.1. Microbial Isolation

Approximately 1.0 g of contaminated soil collected from a petroleum filling station located at King Fahd University of Petroleum and Minerals (Dhahran, Saudi Arabia) was subjected to enrichment cultures in Bushnell–Hass (BH) medium, which consisted of (per liter) (NH_4_)_2_SO_4_ (2.38 g), KH_2_PO_4_ (1.36 g), CaCl_2_·7H_2_O (10.69 g), MgSO_4_·7H_2_O (0.25 g), Na_2_HPO_4_ (1.42 g), FeSO_4_·7H_2_O (0.28 mg), and NaCl (0.50 g) supplemented with BaP (0.05% (*w*/*v*), 2.0 mmol·L^−1^) as the sole carbon source. BaP was directly added to the medium as a powder. This medium was named BH-BaP. Enrichment experiments, which resulted in the isolation of BaP-degrading bacteria, were initiated in a 150 mL flask containing 50 mL of culture medium at 37 °C and 120 rpm. After 3–4 weeks, 1.0 mL of the culture was transferred to fresh culture medium (1/10, *v*/*v*), and the process was repeated 4–5 times until bacterial growth was observed. To separate and culture individual colonies, each culture was transferred to a solid agar plate, which was prepared in rich LB medium containing agar (1%, *w*/*v*). Thereafter, 1.0 mL of each bacterial culture was spread on solid plates and incubated at 37 °C for 24 h. The resulting individual colonies were cultured in 10 mL of LB medium for another 24 h and centrifuged at 3500× *g* for 5 min at 4 °C. The resulting pellet was then cryopreserved in aqueous solution containing glycerol (15%, *v*/*v*).

### 2.2. Bacterial Enumeration

Bacteria were counted following 10-, 100-, and 1000-fold serial dilution in BH medium. Afterward, 1.0 mL of the diluted culture was spread onto solid agar plates, which were incubated at 37 °C for 24 h. Thereafter, the number of individual colonies was counted and presented as colony-forming units per ml (CFU·mL^−1^).

### 2.3. Scanning Electron Microscopy (SEM)

Approximately 100 µL of an aliquot of bacterial cells was cultured in 10 mL of LB medium at 37 °C in a shaker at 37 °C and 120 rpm. After 12 h, bacterial cells were isolated by centrifugation at 3500× *g* for 5 min at 4 °C and suspended in 1 mL of phosphate buffer (PBS, pH7). Approximately 20 µL of a glutaraldehyde solution at a concentration of 0.2% was added to the suspension and incubated for 12 h at room temperature. Thereafter, 100 µL of the suspension was spread on microscopic slides for another 12 h at 37 °C to let glutaraldehyde evaporate. Bacterial cells on the slides were fixed by serial dehydration using ethanol (30%, 50%, 70%, 80%, 90%, and 95% *v*/*v*), followed by coating of the samples with gold. SEM observation was conducted using a JSM-T300 electron microscope (JEOL, Tokyo, Japan).

### 2.4. Species Identification

To identify species, bacterial cells were precultured in 100 mL of rich LB medium and separated by centrifugation at 5000× *g* for 5 min at 4 °C. The resulting cells were lysed, and DNA was extracted and purified using a Qiagen Powerfecal Kit (Qiagen, Hilden, Germany). Thereafter, 16S rRNA gene amplification was performed, followed by sequencing, as reported elsewhere [12]. Briefly, the primers 27F (AGAGTTTGATCMTGGCTCAG) and 1492R (CGGTTACCTTGTTACGACTT) were used to amplify 1400 base pairs (bp) of 16S rRNA genes [13]. Thereafter, this amplicon was sequenced using a Big Dye terminator cycle sequencing kit (Applied Biosystems, Thermo Fisher Scientific, Waltham, MA, USA), the primers 518F (CCAGCAGCCGCGGTAATACG) and 518R (GTATTACCGCGGCTGCTGG), and an automated DNA sequencer (model 3730XL, Applied Biosystems). The bioinformatics basic local alignment search tool (BLAST) from the US National Center for Biotechnology Information (NCBI) databases, along with EzBioCloud, a newly developed taxonomically united database of 16S rRNA gene sequences, was employed to identify and ascertain species [14].

### 2.5. Assessment of Bacterial Degradation of BaP and Other PAHs and the Effects of pH and Salinity

Biodegradation experiments on various PAHs were conducted using the bacterial consortium or individual strains that were cultured in 10 mL of rich LB medium for 24 h. Thereafter, a bacterial load of approximately 5 × 10^4^ CFU·mL^−1^ was cultured in 50 mL of BH medium containing PAHs, which were dissolved in DMSO prior to their addition to BH culture. The biodegradation of BaP, pyrene, phenanthrene, naphthalene, and anthracene was assessed. The biodegradation of salicylic acid and catechol, which are polar substrates, was also studied. These substrates were dissolved directly in the medium. The effects of salinity, temperature, and pH on the ability of the strains (as a consortium or single strains) to grow in the presence of BaP were also assessed at salinity levels of 0, 5, 10, 15, and 20% NaCl (*w*/*v*); temperatures of 30 °C, 35 °C, 40 °C, and 45 °C; and pH values of 5, 6, 7, 8, and 9. All experiments were performed in duplicate.

Bacterial growth was quantified by assessing the doubling time (dt) by fitting growth curves to the equation Q_t_ = Q_o_e^−*k*t^, where Q_t_ and Q_o_ represent the bacterial count at time t and time 0, respectively, and *k* represents the rate of growth. Thereafter, dt (in hours) was calculated using the equation dt = ln(2)/*k*.

### 2.6. BaP Quantification and Substrate Utilization

BaP was quantified using a 100 mL culture of either the bacterial consortium or single bacterial strains in the presence of 20 µmol·L^−1^ BaP over a period of 30 days using a previously reported protocol [15]. For each time point (days 5, 15, and 30), a whole culture sample (100 mL) was sonicated for 30 min and subjected to extraction with ethyl acetate (50 mL × 2). After drying with calcium chloride, the combined organic layers were dried under vacuum and dissolved in 500 μL of chloroform before analysis by GC. The equation Q_t_ = Q_o_e^−*k*t^ was employed to estimate the constant k (Lu et al., 2014). These experiments were performed in triplicate. The following method was used in the GC (using an HP-5 column (30 m, internal diameter = 0.320 mm)): initial oven temperature of 120 °C for 2 min, increased to 250 °C at 11 °C/min, and then held for 50 min; injector temperature, 310 °C; detector temperature, 320 °C; and helium flow rate, 15 mL/min. The injected volume was 1.0 µL, and the split ratio was 10:1.

### 2.7. Quantification of BaP Utilization in Sandy Soil Samples

The ability of bacteria to degrade BaP in the context of the contamination of sand was assessed by spiking BaP (40 µmol) into 100 g of sand in a 5 × 5 cm^2^ Petri dish. Thereafter, precultured bacteria (1 × 10^7^ CFU) were added in each Petri dish with a minimum amount of BH medium to maintain moisture in the samples. Control experiments were prepared with autoclaved bacteria. On days 0, 15, 40, 60, and 90, each Petri dish sample was extracted with ethyl acetate, and the remaining BaP was quantified by GC, as explained in the previous section. These experiments were performed in triplicate.

### 2.8. Identification of Metabolites by HPLC-MS/MS

Individual bacterial strains or their consortium were cultured in 1.0 L of medium in the presence of BaP (4.0 mmol·L^−1^). After 15 days, each culture was extracted with ethyl acetate (300 mL × 2), and the combined organic layer for each strain or the consortium was concentrated to around 200 mL and extracted with sodium hydroxide solution (1.0 M, 200 mL × 2). The aqueous layer was then neutralized with concentrated hydrochloric acid prior to extraction with ethyl acetate (150 mL × 2). The combined organic layer was dried with sodium sulfate, and the solvent was then evaporated under vacuum. The remaining residue was dissolved in 2 mL of 30% (*v*/*v*, methanol:water) prior to analysis using a Shimadzu Nexera Prominence liquid chromatography system interfaced with a mass spectrometer (AB SCIEX X500 QTOF). Liquid chromatography separation was performed using an HC-C_18_ column (4.6 × 250 mm, Agilent, Santa Clara, CA, USA). The mobile phase was set at a flow rate of 1.0 mL·min^−1^. In the gradient program, the metabolites were eluted by a linear gradient that was started at 50% (*v*/*v*, methanol/water), increased to 95% (methanol in water) for 60 min, and decreased to 50% for 4.9 min, for a total run time of 65 min. The parameters of the ion source were a temperature of 300 °C, a curtain gas pressure of 30 psi, a source gas 1 pressure of 160 psi, and an ion source gas 2 pressure of 60 psi. Both positive and negative time-of-flight (TOF)-MS/MS scan modes were applied to enhance the chances of observing potential metabolites. In the negative scan mode, the typical TOF-MS/MS parameters were as follow: ion spray voltage, −4500 V; CAD gas, 7; TOF mass range, 50–1000 Da; accumulation time, 0.15 s; declustering potential, −60 V; declustering potential spread, 0 V; collision energy, 10 V; and collision energy spread, 0 V. The same parameters were used in the positive scan mode, excluding differences in the ion spray voltage (5500 V), declustering potential (60 V), and collision energy (10 V).

### 2.9. Statistical Analyses

One-way analysis of variance (ANOVA), Student’s *t*-test, and a linear regression fitting model were used to analyze the data. These analyses were performed using the well-established contributed packages in R software using the “aov”, “t.test”, and “lm” functions in the “stat” package for one-way ANOVA, Student’s *t*-test, and linear regression analysis, respectively [16]. Data strength for linearity was established using Pearson’s correlation coefficient, and the level of significance in all tests was fixed at *p* < 0.05.

## 3. Results and Discussion

### 3.1. Bacterial Isolation

Enrichment experiments using soil containing BaP as the sole carbon source led to the isolation of a bacterial consortium. Furthermore, the use of solid culture on agar plates permitted the identification of three distinct colonies: JBZ1A, JBZ2B, and JBZ5E. Under light microscopy (×40–1000), JBZ1A colonies were punctiform, rod-shaped, and flat with intact margins. JBZ2B colonies were spherical, yellow, circular, and raised with entire margins. Finally, JBZ5E colonies were creamy, circular, and convex with intact margins. JBZ2B and JBZ5E were Gram-positive, whereas JBZ1A was Gram-negative. SEM confirmed that JBZ1A and JBZ5E were rods with sizes of 0.2 × 1.5 and 0.3 × 2.0 µm^2^, respectively, whereas JB2B was a coccus with a diameter of approximately 0.7 µm (Appendix A).

### 3.2. Species Identification

16S rRNA gene amplification generated fragments of approximately 1200 bp in size, and subsequent sequencing and comparison of their 16S rRNA gene sequences using BLAST led to the identification of JBZ1A as *B. japonicum* (NCBI, accession number, GI: 1788953717), JBZ2B as *M. luteus* (GI: 1788953718), and JBZ5E as *B. cereus* (GI: 1788953719) based on 99% homology. The use of the EzBioCloud database yielded the same names of bacterial species.

Bacteria belonging to the genus *Bradyrhizobium*, which are symbiotic N_2_-fixing bacteria, have been studied extensively in the context of N_2_ fixation [17]. However, little has been reported concerning the biodegradation of aromatic pollutants. Strains of *B. japonicum* have been revealed to degrade the monoaromatic hydrocarbons 5-nitroanthranilic acid, vanillic acid, 4-hydroxybenzoic acid, and protocatechuic acid [18,19,20]. To the best of our knowledge, the current report is the first to describe the degradation of a PAH by a *Bradyrhizobium* bacterial strain.

A recent report revealed the ability of a strain of *M. luteus* to degrade PAHs, including BaP [21]. However, in this investigation, crude oil was used as the carbon source; therefore, it is possible that the degradation of PAHs, including BaP, could have been the result of cometabolism [11]. A bacterial strain belonging to the *Micrococcus* genus was demonstrated to degrade naphthalene, a two-ring PAH, along with its derivatives, including the insecticide carbaryl [22]. Degradation of the aromatic biphenyl and some of its derivatives has also been reported for *Micrococcus* bacterial strains [23,24]. Likewise, phenanthrene and pyrene degradation has been demonstrated in *Micrococcus* spp. and *M. luteus* strains, respectively [25,26]. The current work provides evidence that *Micrococcus* spp. can also degrade BaP.

Several studies have reported the degradation of BaP by *Bacillus* spp., including *B. cereus.* For instance, a consortium of *B. cereus* and *B. vireti* was reported to degrade BaP [27]. Interestingly, BaP degradation has been reported in thermophilic strains of *Bacillus*, namely *B. licheniformis* and *B. subtilis* [28,29,30]. Degradation of other aromatic compounds has also been reported in *Bacillus* strains, including *B. cereus* [31,32,33,34,35]. This work confirmed the ability of *B. cereus* to degrade BaP when used as a sole carbon source.

### 3.3. Effects of pH, Temperature, and Salinity on Bacterial Growth

Before initiating these studies, given that BaP, along other PAHs, was dissolved in DMSO, as mentioned in Material and Methods, the effect of DMSO (0.01% *v*/*v*) on bacterial growth was assessed, and the results showed none of the tested bacteria species grew for 60 days. To establish the optimal growth conditions for the three strains and their consortium in the presence of BaP as the sole carbon source, dt was assessed by varying pH, temperature, and salinity. In the analysis, pH was varied from 5 to 9, with the temperature and salinity fixed at 37 °C and 0.5% NaCl (*w*/*v*), respectively. As presented in Table 1, no growth was observed at pH 5 for *B. cereus* or at pH 9 for *B. japonicum* and *M. luteus*. The optimal pH, as indicated by the lowest dt, was pH 7 for *B. cereus*, *M. luteus*, and the consortium and pH 6 for *B. japonicum.* The lowest growth was associated with pH 9; however, the difference in dt was significant at pH 9 versus the other pH values only for the consortium. The dt of *B. cereus* at pH 5 was statistically significantly different from that at pH 7. Overall, the pH range of 6–8 was associated with better growth for these strains (either alone or as a consortium).

Growth was also assessed at 30 °C, 35 °C, 37 °C, 40 °C, and 45 °C, with salinity and pH fixed at 0.5% (*w*/*v*) and 7, respectively. Overall, the differences in dt were not significant at 30–40 °C for single strains or the consortium, and the values ranged from 20 to 34 h. However, *M. luteus* did not grow at 45 °C, and higher dt values (51–53 h) were observed for the two remaining strains and the consortium (Table 1). The differences in dt at 45 °C were statistically significant relative to those at other temperatures for *B. cereus* and *B. japonicum* (Table 1).

Finally, the effect of salinity was tested with 0.5, 2, 4, and 8% NaCl (*w*/*v*) by fixing the temperature at 37 °C and the pH at 7. Neither the tested strains nor the consortium grew in the presence of 6% or 8% NaCl, and in the case of *M. luteus*, no growth was observed, even at 4% NaCl. Overall, dt increased with increasing salinity and was lowest at 0.5% NaCl; however, the differences in dt in the presence of 0.5% NaCl versus that in the presence of 2% or 4% NaCl were statistically significant only in *B. cereus* (Table 1).

All strains were enriched and isolated at pH 7, 37 °C, and 0.5% NaCl. Thus, it is not surprising that the aforementioned values were optimal for growth of the consortium. Significantly reduced growth was observed at pH >8, >4% NaCl, and >40 °C. However, few bacterial strains that can degrade BaP at high temperatures (50–70 °C) have been reported, such as strains of *B. subtilis*, *Bacillus* spp., *Thermus* sp., *and B. licheniformis* [28,29,30]. Concerning salinity, the degradation of BaP by halophilic bacteria has been reported in the presence of moderate salinity (3% NaCl) by strains of *Enterobacter cloacae, Stenotrophomonas maltophilia*, and *Ochrobactrum* sp. [36,37]. To the best of our knowledge, no bacterial strain capable of degrading BaP under acidic and alkaline conditions has been reported to date.

### 3.4. Growth of the Bacterial Consortium in Comparison with the Individual Strains in the Presence of Increasing BaP Concentrations

The ability of individual strains and the bacterial consortium to grow in the presence of various concentrations of BaP (4, 20, 40, 100, 200, and 400 µmol·L^−1^) was assessed at pH 7 and 37 °C (4 µmol·L^−1^ BaP corresponds to approximately 1 ppm). The results are summarized in Figure 1; overall, the data illustrate that bacterial growth for both individual strains and the consortium decreased with increasing BaP concentrations.

For instance, for the consortium, the dt increased from 15 h at the lowest concentration to 31 h at the highest concentration. Likewise, for individual strains, the ranges of dt were 22–30, 21–30, and 26–71 h for *B. cereus*, *M. luteus*, and *B. japonicum*, respectively. These findings also indicate that the bacterial consortium had a higher growth rate than the individual strains at all tested concentrations, except that at the highest BaP concentration (400 µmol·L^−1^), the consortium had a similar dt to that of *B. cereus* and *M. luteus*. At this concentration, *B. japonicum* harbored a higher dt (>70 h). At the lowest BaP concentration (4 µmol·L^−1^), the consortium had a dt of 15 h, but the corresponding values for individual bacterial strains were >22 h. ANOVA revealed a significant correlation between the BaP concentration and dt (*p* < 0.05) for all strains, excluding *M. luteus*. Furthermore, trend analysis illustrated that these relationships followed a linear equation (dt = 0.15C + 14.59, (R^2^ = 0.92, where C is the BaP concentration) for the consortium; dt = 0.49C + 24.65, (R^2^ = 0.82) for *B. japonicum*; dt = 0.08C + 22.72, (R^2^ = 0.6) for *M. luteus*, and 0.09C + 23.40, (R^2^ = 0.68) for *B. cereus*), and these trends were statically significant (*p* < 0.001). PAHs in general and BaP in particular have been demonstrated to be toxic to cells, including bacteria. Therefore, it is not surprising that, in general, an increase in the BaP concentration leads to a decrease in bacterial growth; this observation has been previously reported for BaP in *Staphylococcus haemolyticus* (Nzila et al., 2020) and for pyrene in *Ochrobactrum* sp. [36], *Halomonas shengliensis*, and *H. smyrnensis* [15]. Similar results were reported in the degradation of LMW-PAHs, anthracene, and phenanthrene in *B. licheniformis, Ralstonia pickettii, Pseudomonas citronellolis*, and *S. maltophilia* [12,38].

### 3.5. Growth in the Presence of Various Aromatics

The ability of individual bacterial strains and the consortium to degrade the PAHs pyrene, phenanthrene, anthracene, and naphthalene, as well as the monoaromatic hydrocarbons salicylic acid and catechol, is summarized in Figure 2.

These experiments were conducted under the conditions of pH 7, 37 °C, and 0.5% NaCl. Overall, the use of BaP and pyrene was associated with higher dt values than the other tested aromatic compounds, and the bacterial consortium had a lower dt than all individual strains. Regarding the consortium, dt was approximately 18 h for both BaP and pyrene, and the values for monoaromatic hydrocarbons ranged 13–14 h, whereas the failures for the three-ring PAHs phenanthrene and anthracene and the two-ring PAH naphthalene ranged from 16 to 18 h. The dt ranges were 14–20, 16–23, and 18–24 h for *B. cereus*, *M. luteus*, and *B. japonicum*, respectively. Although the differences in dt were not statistically significant (ANOVA, *p* > 0.05), these results are supported by many previous observations that the difficulty of biodegradation increases with increasing complexity among PAHs, and because bacteria degrade PAHs in a stepwise ring-opening process, a bacterium that degrades PAHs with numerous rings can also degrade PAHs with fewer rings. For instance, efficient degradation of PAHs with lower molecular weights than BaP has also been previously reported for the following BaP-degrading bacteria: *Ochrobactrum* sp. BAP5 [39], *Ochrobactrum* sp. VA1 [36], *Hydrogenophaga* sp. PYR1 [40], *Cellulosimicrobium cellulans* CWS2 [41], *Rhizobium tropici* CIAT 899 [42], *Klebsiella pneumonia* PL1 [43], and *Pseudomonas* sp. JP1 [44].

### 3.6. Quantification of BaP Utilization in BH Medium and Sandy Soil Samples

The rate at which each bacterial strain degraded 40 µmol·L^−1^ BaP in BH medium at pH 7 and 37 °C in comparison with the consortium is summarized in Figure 3.

Overall, the consortium had the highest rate of degradation, removing almost 60% of BaP within the first 15 days, whereas less than 20% of BaP remained in the culture medium on day 30. Regarding the individual strains, none had degraded more than 50% of BaP by day 5, and on day 30, *B. cereus*, *M. luteus*, and *B. japonicum* had degraded approximately 75%, 44%, and 47% of BaP, respectively. The degradation rates were 1.06 (R^2^ = 0.98), 0.90 (R^2^ = 0.95), 0.38 (R^2^ = 0.89), and 0.66 µmol·L^−1^·day^−1^ (R^2^ = 0.96) for the consortium, *B. cereus*, *M. luteus*, and *B. japonicum*, respectively. Based on these results, *B. cereus* made the strongest contribution to degradation activity in the consortium, followed by *B. japonicum* and *M. luteus*.

In a recent report, Nzila et al. [45] compared the degradation rates of BaP in various studies. Overall, degradation rates were reported to be lower for single bacterial strains than for consortia, in line with the observations in the current work and those reported elsewhere [46]. In addition, this comparative analysis illustrated that the degradation rates ranged from 0.03 to 2.4 µmol·L^−1^·day^−1^ in minimum mineral medium, in agreement with the values obtained in the current work (0.38–1.06 µmol·L^−1^·day^−1^).

BaP degradation by the consortium was also tested in sandy soil spiked with BaP (40 µmol·kg^−1^). As presented in Figure 4, the consortium degraded BaP under this condition; however, this degradation occurred at a lower rate than that in BH medium.

After 90 days, less than 35% of BaP was degraded, whereas 90% of BaP (40 µmol·L^−1^) was degraded in BH medium (as discussed in the previous section). The degradation rate in sandy soil spiked with BaP was almost 0.01 µmol·kg^−1^·day^−1^ (R^2^ = 0.98) versus 1.06 µmol·L^−1^·day^−1^ in BH medium.

Bacterial bioaugmentation of soil samples to increase the biodegradation of PAHs has been attempted, with inconsistent results. For instance, single strains or microbial consortia were assessed for the degradation of phenanthrene [47], light and heavy fractions of petroleum products [48], and diesel [49]. However, using microbial consortia consisting of three bacterial strains selected following enrichment in pyrene revealed no benefit of bioaugmentation in contaminated forest soil samples [50]. In another study, the use of a consortium of PAH-degrading bacteria (primarily consisting of *Actinomycetes*) failed to increase the degradation of PAHs in soil samples [51]. Similar results were obtained for the bioaugmentation of slurry-phase bioreactors by *Paenibacillus validus* PR-Pl, *Sphingomonas* spp. PR-PI2, and *Arthrobacter* spp. concerning the degradation of PAHs [52]. Several factors can explain these contradictory results, including the soil quality, the bacterial load, the ability of these added bacteria to grow in the soil, and the pre-existence of bacteria capable of depredating PAHs in the soil [50,51,52,53,54]. It was recommended that the growth of bacteria to be added to the soil should be assessed prior to bioaugmentation experiments. The current study revealed a significant BaP degradation rate in soil samples, highlighting the potential of this consortium to remove BaP from contaminated environments.

### 3.7. Metabolite Identification

To identify metabolites involved in BaP degradation by the three stains and the consortium, reversed-phase HPLC-MS/MS was conducted (Table 2). The use of *B. cereus* led to the identification of a metabolite with the molecular formula C_20_H_12_O_2_ and [M + 1]^+^ at *m*/*z* 285.091; this metabolite was identified as a dihydroxy-BaP isomer. A compound with [M + 1]^+^ at *m*/*z* 301.1415 was observed in individual strains and the consortium. This compound had the molecular formula of C_21_H_16_O_2_, and it was identified as a methylated dihydrodiol-BaP isomer—another confirmation of the presence of dihydroxy-BaP metabolites.

Dihydroxy-BaPs and their dihydrodiol-BaPs have been reported in several bacterial species, including *Beijerinckia* B-836, *Mycobacterium* RJGII-135s, *M. vanbaalenii* PYR-1, and *S. haemolyticus* [45,55,56,57]. These metabolites (dihydroxy-BaP and dihydrodiol-BaP or their isomers) are generated by the action of dioxygenases under aerobic conditions, leading to the addition of hydroxyl groups at positions C4 and C5, C7 and C8, C9 and C10, or C11 and C12 (see the BaP structure in Figure 5) [11].

However, the results obtained by MS do not allow for the identification of specific isomers. Studies involving the use of ^14^C-labeled BaP could help to identify specific isomers [58].

Another metabolite produced by the consortium had [M + 1]^+^ at *m*/*z* 301.0761, corresponding to the molecular formula C_20_H_12_O_3_, which was identified as 4-formylchrysene-5-carboxylic acid. This supports the occurrence of ring cleavage at C4 and C5, suggesting that the detected dihydroxy-BaP could be 4,5-dihydroxy-BaP (Appendix A). This metabolite was previously observed in BaP degradation using *Mycobacterium* sp. RJGII-135 and *S. haemolyticus* [45,56]. Although a 4-formylchrysene-5-carboxylic acid metabolite was identified using the consortium, we cannot assign it any individual strain. Nevertheless, based on the aforementioned information, a tentative degradation pathway of BaP is presented in Figure 5. As stated previously, the action of dioxygenase leads to the formation of dihydrodiol-BaP, which is converted to dihydroxy-BaP through dehydrogenation. Thereafter, ring cleavage at C4–C5 occurs to generate formylchrysene-5-carboxylic acid, and subsequent reactions lead to intermediates involved in the Krebs cycle. The intermediate dihydrodiol-BaP can also be methylated to generate methylated dihydrodiol-BaP (Figure 5).

## 4. Conclusions

Three bacterial strains capable of degrading BaP, namely *B. japonicum*, *M. luteus*, and *B. cereus*, were isolated and characterized. These strains were also demonstrated to degrade PAHs with lower molecular weights than BaP, along with monoaromatic hydrocarbons. The consortium of the three strains was more active in degrading BaP than individual strains. Moreover, this consortium could degrade BaP spiked with soil, although this degradation rate was lower than that of BaP in culture medium. HPLC-MS/MS permitted the detection of dihydroxy-BaP and 4-formylchrysene-5-carboxylic acid, and a tentative degradation pathway was proposed.

## Figures and Tables

**Figure 1 ijerph-20-01855-f001:**
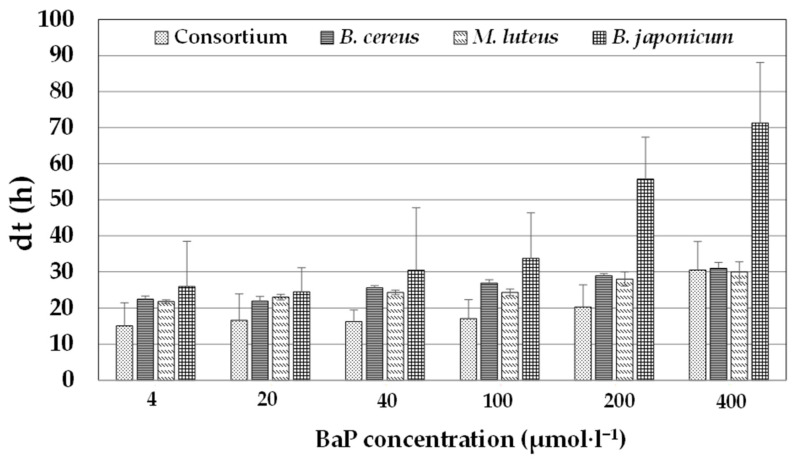
Bacterial growth in the presence of BaP concentrations. Doubling time (dt) of single cultures of the strains *B. japonicum*, *M. leteus*, and *B. cereus* and of a consortium of these bacteria in the presence of various concentrations of benzo[a]pyrene (BaP).

**Figure 2 ijerph-20-01855-f002:**
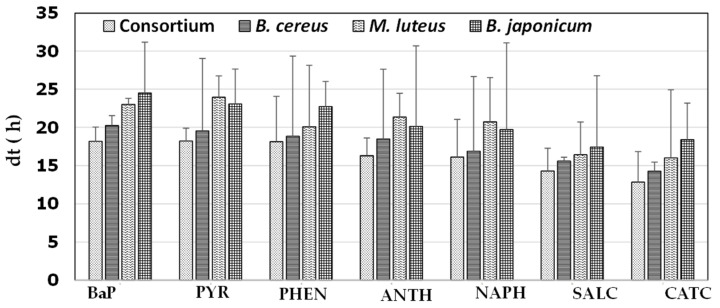
Bacterial growth in the presence of various aromatics. Doubling time (dt) of individual bacterial strains and the consortium in the presence of the aromatic compounds benzo[a]pyrene (BaP), pyrene (PYR), phenanthrene (PHEN), anthracene (ANTH), naphthalene (NAPH), salicylic acid (SALC), and catechol (CATC).

**Figure 3 ijerph-20-01855-f003:**
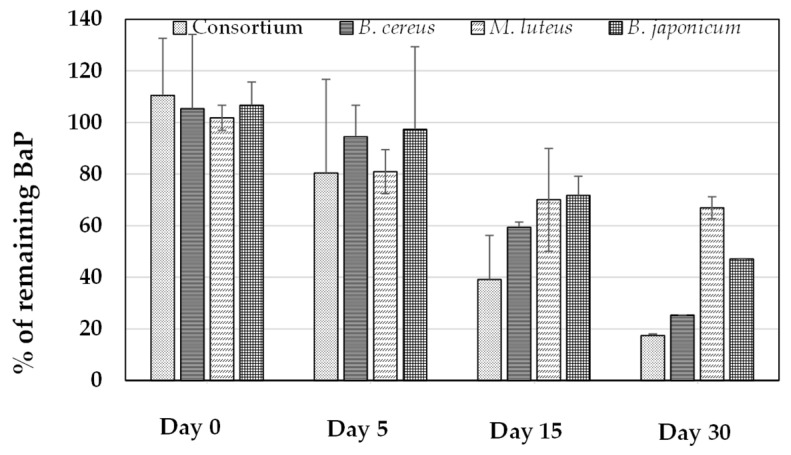
Degradation rates of individual bacteria and the consortium in the presence of 40 µmol·L^−1^ benzo[a]pyrene (BaP) in Bushnell–Hass medium at pH 7 and 37 °C.

**Figure 4 ijerph-20-01855-f004:**
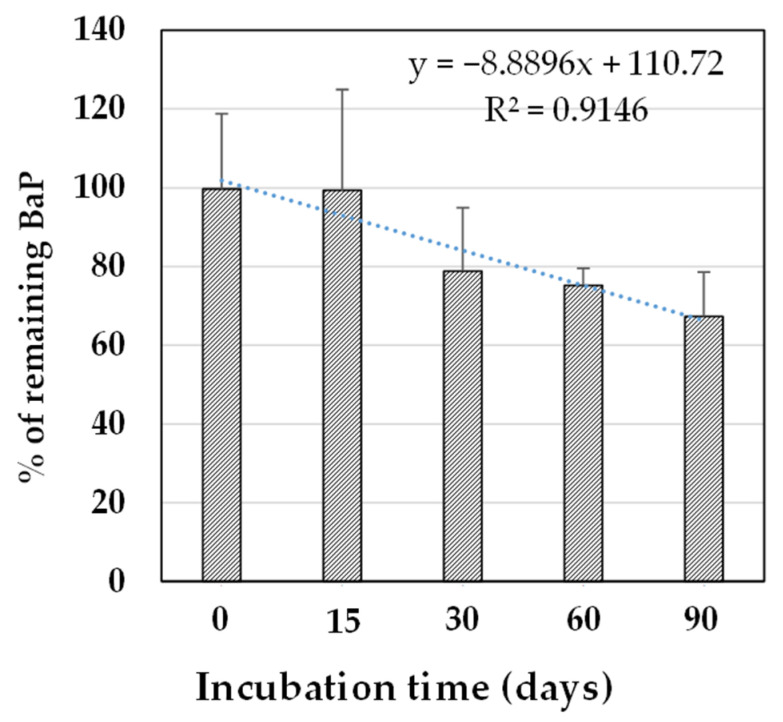
Degradation rate of the consortium in sandy soil spiked with 40 µmol·kg^−1^ benzo[a]pyrene (BaP).

**Figure 5 ijerph-20-01855-f005:**
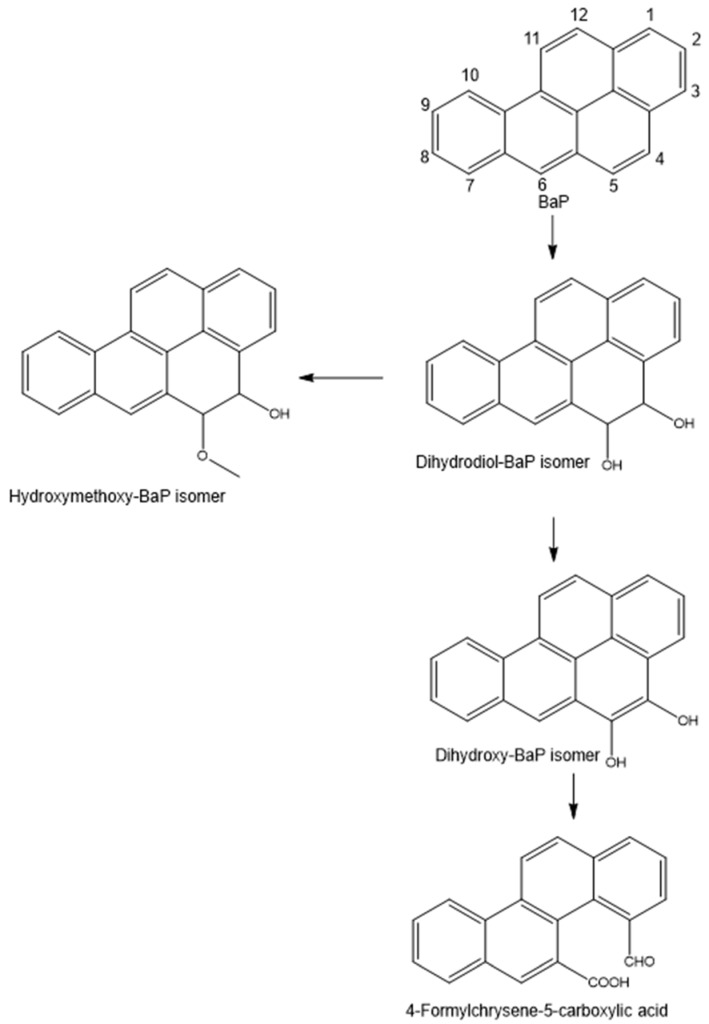
Proposed degradation pathways of benzo[a]pyrene (BaP) by *Bradyrhizobium japonicum*, *Micrococcus luteus*, and *Bacillus cereus*.

**Table 1 ijerph-20-01855-t001:** Doubling time (dt) for the degradation of benzo[a]pyrene by *B. cereus*, *B. japonicum*, *M. luteus*, and a consortium of these three bacterial strains as a function of pH, temperature, and salinity.

				dt (h)	
Condition		*Consortium*	*B. cereus*	*B. japonicum*	*M. luteus*
pH	5	30.3 ± 4.2 ^a^*	30.3 ± 3.0 ^a^	ND	29.8 ± 2.3
	6	24.6 ± 7.2 ^b^	23.3 ± 1.9	23.3 ± 1.9	26.3 ± 2.0
	7	18.1 ± 1.9 ^c^	20.2 ± 1.3 ^a^	33.8 ± 3.4	23.0 ± 1.6
	8	35.1 ± 7.7 ^d^	27.1 ± 0.6	24.5 ± 5.9	31.2 ± 2.9
	9	63.3 ± 8.4 ^a,b,c,d^	ND	62.9 ± 41.8	ND
Temperature (°C)	30	22.2 ± 1.4	25.2 ± 3.8 ^a^	29.1 ± 5.0 ^a^	24.7 ± 1.3
	35	20.4 ± 1.5	21.2 ± 3.7 ^b^	22.2 ± 1.3 ^b^	22.9 ± 4.4
	37	18.1 ± 1.9	20.2 ± 1.7 ^c^	24.5 ± 6.0 ^c^	23.0 ± 1.6
	40	23.0 ± 6.7	23.4 ± 1.9 ^d^	34.2 ± 4.0	26.3 ± 2.0
	45	53.2 ± 19.4	52.1 ± 9.0 ^a,b,c,d^	51.8 ± 7.9 ^a,b,c^	ND
Salinity (% NaCl)	0.5	18.1 ± 1.9	20.2 ± 1.5 ^a^	24.5 ± 6.0	23.0 ± 1.6
	2	24.2 ± 0.5	22.5 ± 1.5 ^b^	30.3 ± 4.2	33.4 ± 7.4
	4	25.9 ± 3.5	30.4 ± 1.5 ^a,b^	37.6 ± 8.8	ND

* Values with the same letters were significantly different (*p* < 0.05).

**Table 2 ijerph-20-01855-t002:** High-resolution mass spectrometry data for the detected benzo[a]pyrene (BaP) metabolites using *Bradyrhizobium japonicun* (JBZ1A)*, M. luteus* (JBZ2B), and *B. cereus* (JBZ5E).

Metabolite	Observed Molecular Ion Mass	Molecular Formula	Calculated Exact Molar Mass	Strain(s) in Which This Metabolite Is Observed
Dihydroxy-BaP	285.091 [M + 1]	C_20_H_12_O_2_	285.091 [M + 1]	JBZ5E
Dihydroxy-BaP	281.247 [M − 1]	C_20_H_10_O_2_	281.060 [M − 1]	JBZ2B, CON
Methylated-dihydrodiol-BaP	299.258 [M − 1]	C_21_H_16_O_2_	299.107 [M − 1]	JBZ1A, JBZ2B, JBZ5E, CON
4-Formylchrysene-5-carboxylic acid	299.259 [M − 1]	C_20_H_12_O_3_	299.071 [M − 1]	CON

CON, consortium.

## Data Availability

Data are available upon request to the authors.

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
