# Peer review of "Benzo[A]Pyrene Biodegradation by Multiple and Individual Mesophilic Bacteria under Axenic Conditions and in Soil Samples"

_ijerph, 2023, doi:10.3390/ijerph20031855_

Round 1

Reviewer 1 Report

Dear Authors, Thank you for studying a crucial environmental problem of BaP biodegradation. I appreciate your attempt of excluding co-metabolism in the pathway of BaP degradation. I found your publication as a very informative, based on standard and contemporary microbiological methods (including genetic identification of bacterial species), with a good interpretation of obtained results and conclusions drawn based on experimental data. I am absolutely convinced that this manuscript is within the scope of the journal and can be published with minor corrections. I feel that your research could open new options in achieving effective environmentally sound methods of biodegradation of so harmful substances as PAHs. Some (in majority just technical and editorial) errors have been marked in the text.

Author Response

We have addressed all the comment you made. More specifiquement, we have the following:

Reviewer 1:

Line 112: Done and through out the text

Figure 1: The format of the figures have been changed to  conform to the MDPI format

Line 316: the capital letters has been removed

Line 346: Error on the paragraph has been corrected

Line 398: Error on the paragraph has been corrected

Line 445: doubling numbering of the reference has been corrected.

Reviewer 2 Report

The study isolated three new bacterial strains which can be applied in the biodegradation pathway in the contaminated soil. The entire study was intact experimented and well designed. The logic was fluent, and the writing is well. The only revised part will be the structure of the manuscript following the journal's requirement. 

Author Response

We thank the reviewer for these remarks about the good quality of the manuscript. 

Regarding the journal format, the same comments have been made by other reviewers, and we have reviewed the all manuscript and made the necessary changes so that it can adhere to the journal requirement. 

Reviewer 3 Report

The study reports the isolation and identification of 3 bacterial species capable of degrading BaP. The study is well-planned and executed and offers new insights related to de biodegradation process of a molecule potentially toxic and harmful to human health and the environment.

It is suggested to review some minor grammatical mistakes and typos through the text.

Author Response

We thank the reviewer for these comments.

We have carefully read the manuscripts and have made the necessary corrections.